# Supervised Knowledge Makes Large Language Models Better In-context Learners

**Linyi Yang**[1,2,*] **Shuibai Zhang**[1*], **Zhuohao Yu**[3*], **Guangsheng Bao**[1], **Yidong Wang**[3],
**Jindong Wang**[4], **Ruochen Xu**[4], **Wei Ye**[3], **Xing Xie**[4], **Weizhu Chen**[4], **Yue Zhang**[1,2,†]

[1]School of Engineering, Westlake University, [2]Westlake Institute for Advanced Study
[3]Peking University, [4]Microsoft

## Abstract

Large Language Models (LLMs) exhibit emerging in-context learning abilities through prompt engineering. The recent progress in large-scale generative models has further expanded their use in real-world language applications. However, the critical challenge of improving the generalizability and factuality of LLMs in natural language understanding and question answering remains under-explored. While previous in-context learning research has focused on enhancing models to adhere to users' specific instructions and quality expectations, and to avoid undesired outputs, little to no work has explored the use of task-Specific fine-tuned Language Models (SLMs) to improve LLMs' in-context learning during the inference stage. Our primary contribution is the establishment of a simple yet effective framework that enhances the reliability of LLMs as it: 1) generalizes out-of-distribution data, 2) elucidates how LLMs benefit from discriminative models, and 3) minimizes hallucinations in generative tasks. Using our proposed plug-in method, enhanced versions of Llama 2 and ChatGPT surpass their original versions regarding generalizability and factuality. We offer a comprehensive suite of resources, including 16 curated datasets, prompts, model checkpoints, and LLM outputs across 9 distinct tasks [1]. Our empirical analysis sheds light on the advantages of incorporating discriminative models into LLMs and highlights the potential of our methodology in fostering more reliable LLMs.

## 1 Introduction

Trained on extensive volumes of data with numerous parameters, large language models (LLMs) have garnered significant performance across diverse tasks. Their in-context learning (ICL) ability positions them as foundational models to adeptly address various downstream tasks, ranging from natural language understanding (Chowdhery et al., 2022; OpenAI, 2023a;b) to reasoning (Wei et al., 2022; O'Brien & Lewis, 2023), and planning (Shen et al., 2023).

Despite their robust performance, LLMs come with their own set of challenges; they demand substantial resources for training and deployment, demonstrate slow inference times, and are susceptible to hallucination (Li et al., 2023a). Conversely, supervised task-specific language models (SLMs) [2] offer cost-efficiency in both training and inference, despite losing general multi-task capacities. Owing to their smaller scale and reduced training cost, SLMs can swiftly adapt to distinct tasks, learning task-specific knowledge (Devlin et al., 2018). As new and tailored tasks constantly emerge in real applications, they can pose out-of-distribution (OOD) challenges to LLMs. It has been shown even with ICL, LLMs generally underperform SLMs in such natural language understanding tasks, with an increased tendency for hallucination when completing classification tasks (Sun et al., 2023b).

---

[*]Equal Contribution.

[†]Correspondence to: zhangyue@westlake.edu.cn

[1]The code and data are released at: https://github.com/YangLinyi/Supervised-Knowledge-Makes-Large-Language-Models-Better-In-context-Learners

[2]SLMs refers to cost-efficient, task-specific, pre-trained discriminative language models in this work.

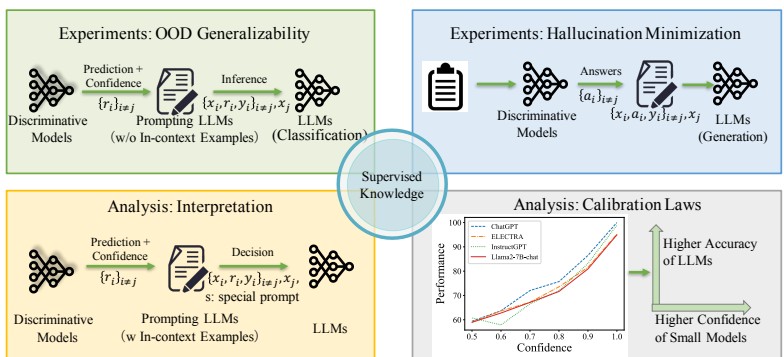

Figure 1: We denote $(x_i, y_i)$ as a question-answer pair and our receipt $r_i$ is inserted between the question-answer pair. Supervised knowledge plays a key role in improving OOD generalizability and factuality of LLMs. While the following two analysis tasks aim to explain why our method outperforms the traditional in-context learning method.

Most of the existing research predominantly segregates LLMs and SLMs as independent learning paradigms (Zhao et al., 2023), overlooking their potential interconnection. Given the distinct advantages and disadvantages of LLMs and SLMs, a fundamental question emerges: *Can SLMs enhance LLMs' performance?* Specifically, can SLMs bolster LLMs' reliability in OOD scenarios while minimizing hallucination? Prior research (Li et al., 2023b) hints at the potential for enhancing the performance of LLMs with the assistance of a smaller task-specific language model, but relatively little work addresses this research question systematically and empirically. To this end, we conduct a set of systematic empirical evaluations. Our assumption is that SLMs and LLMs have underlying complementarity in terms of knowledge – while SLMs are equipped with *task* knowledge thanks to supervised training data, LLMs are endowed with rich *domain* knowledge from large-scale pretraining. Consequently, we focus on OOD settings of various tasks in our evaluation.

This paper introduces **SuperContext**, a versatile and straightforward in-context learning strategy to harness the strength of small models to augment LLMs, particularly focusing on OOD generalization and factuality. At the heart of SuperContext is the integration of SLM outputs representing the **supervised knowledge** into LLM prompts, exemplified by incorporating the predictive results and confidence of a discriminative model during the LLM's inference stage. This idea is similar in spirit to existing work on retrieving information from external knowledge bases or API tools, such as unstructured corpora, structured databases, Wikipedia, and Google API (Borgeaud et al., 2022; Larson et al., 2022; Li et al., 2023c). However, since our goal is to allow reliable *task adaptation* rather than *knowledge acquisition*, the consulting agent becomes SLMs rather than search engines.

SuperContext is examined in two experiments and two perspectives of analysis. The first task is OOD natural language understanding (NLU), where LLMs are enhanced with the supervised knowledge from task-specific fine-tuned models for OOD datasets. The discriminative model is fine-tuned on task-specific data from diverse domains, and seamlessly bridges the gap between the extensive pre-trained model and task-specific data, eliminating overfitting. The second task is question answering containing unanswerable questions, where we underscore SuperContext capability to curtail hallucinations, addressing them through a discriminative-model-enhanced approach. To analyze the underlying mechanisms, an interpreter is constructed to elucidate why SuperContext transcends traditional in-context learning methods, based on a comprehensive post-hoc analysis. In addition, extensive quantitative and qualitative assessments delve into how small models facilitate LLMs in tackling the classification conundrum.

We conduct experiments on both zero-shot and few-shot settings of natural language understanding and question answering (QA). SuperContext is validated on a comprehensive OOD benchmarks GLUE-X (Yang et al., 2022), and a QA dataset, SQuAD 2.0 (Rajpurkar et al., 2018). Empirical results show that our method significantly outperforms LLMs and SLMs with both zero-shot and few-shot settings on 9 distinct tasks using the OOD setting we consider. To the best of our knowledge, this work propounds SuperContext as a pioneering approach to systematically integrate SLMs into LLM inference decisions, significantly enhancing LLM performance, especially in managing OOD data and mitigating hallucinations, thereby contributing to the advancement of more generalizable and factual deployment of LLMs.

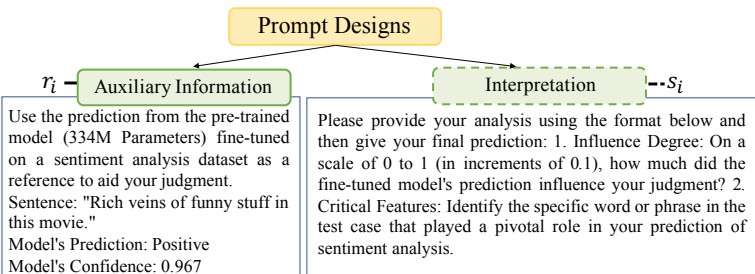

Figure 2: Illustration of prompt designs, where the supervised knowledge provided by the discriminative model is defined as $r_i$, and the optional interpretation prompt is denoted as $s_i$.

## 2 METHOD

### 2.1 IN-CONTEXT LEARNING BASELINE

**In-context learning** (ICL) has become the cornerstone of stimulating the ability of large language models (LLMs) (Dong et al., 2022). To facilitate the evaluation of the traditional in-context learning and our method, in-domain data is provided for several NLU tasks, with each task consisting of 16-shot examples. Denote $(x_i, y_i)$ as a question-answer pair and $S_j$ is the index set of in-context learning samples where $n = |S_j|$ is the number of shots. The few-shot examples are denoted as $\{x_i, y_i\}_{i \in S_j \subset [1,N] \backslash \{j\}}$, where $i \in [1..N]$ and $N$ is the number of problem instances for the task. Formally, traditional in-context learning is based on the following assumption (Xu et al., 2023b):

$$p_{LLM}\left(y_j \mid \{x_i, y_i\}_{i \neq j}, x_j\right) \approx p_{LLM}\left(y_j \mid \{x_i, y_i\}_{i \in S_j}, x_j\right), \quad \forall S_j \subset [1, N] \backslash \{j\}. \quad (1)$$

In a nutshell, Eq. (1) indicates that the probability $p_{LLM}\left(y_j \mid \{x_i, y_i\}_{i \in S_j}, x_j\right)$ of a given LLM generating the response $y_j$ when prompted with the concatenation of the few-shot examples with the discriminative model's prediction, confidence, and the special prompt $s_i$ is approximately invariant to the exact choice of the few-shot examples. We consider both zero-shot and few-shot settings in this work. Notably, the choice and even the order of the examples can have a substantial impact on the test performance (Lu et al., 2021). To mitigate such impact, we employ a thrice resampling with the replacement method for computing the average results.

The key to designing alternatives for ICL is to find the appropriate knowledge elsewhere to embed into the decoding process of the LLM. Recently, Li et al. (2023b) proposed the contrastive decoding approach that exploits the contrasts between the expert and amateur language models of different sizes by choosing tokens that maximize their log-likelihood difference. Their approach generates high-quality texts with the help of an amateur model. However, their approach still requires performing contrastive mapping between those two models in training, which could be tedious. In contrast to their work, the central question that we address is: *"Can we develop a cheap and generalized in-context learning approach that can serve more tasks?"*

### 2.2 SUPERCONTEXT

We propose **SuperContext**, a simple and general approach for in-context learning that incorporates the auxiliary knowledge from a small, discriminative model with LLMs when making predictions for new tasks. This is accomplished through the integration of instruction and the prediction derived from a fine-tuned (small) discriminative language model. Specifically, our receipt $r_i$ is inserted between the question-answer pair: $\{x_i, r_i, y_i\}$. In our work, $r_i$ plays two roles: 1) it provides the discriminative model's prediction and confidence; 2) it further explains the prediction from two aspects, questioning LLMs to answer it learns from which in-context example and which kind of rationale is important.

As shown in Figure 2, we take the sentiment analysis (SST-2) task as an example to illustrate the prompt design. Throughout the process, we do not use any labels from corpus Y as demonstration examples, which aligns with the scenarios in the real world, as typical data points are OOD for the model. In particular, the training set and in-context examples are both drawn from the in-domain dataset, while the training set is used to fine-tune the SLM and in-context examples are used as

Table 1: Data statistics of SuperContext, which describes the source and size for OOD tests of NLU and hold-out test of QA.

| ID | SST-2 | MNLI | QNLI | RTE | MRPC | QQP | STS-B | CoLA | SQuAD 2.0 |
|---|---|---|---|---|---|---|---|---|---|
| OOD | IMDB Yelp Amazon Flipkart | MNLI-mis SNLI | NewsQA | SciTail HANS | QQP Twitter | MRPC Twitter | SICK | Textbook | Train: 130,319 Dev:11,873 |

the prompt. The interpretation prompt $s_i$ is an optional component in SuperContext that should be inserted between the input prompt and test example, where the output is expected to include: 1) the index of influential in-context examples; and 2) the rationale used when making the prediction.

Formally, SuperContext is based on the following assumption:

$$p_{LLM}\left(r_i, y_j \mid \{x_i, r_i, y_i\}_{i \neq j}, x_j, s_i\right) \approx p_{LLM}\left(r_i, y_j \mid \{x_i, r_i, y_i\}_{i \in S_j}, x_j, s_i\right), \qquad (2)$$

where our method can be represented as $\{x_i, r_i, y_i\}_{i \in S_j \subset [1,N] \setminus \{j\}}$ of given LLM, where $i \in [1..N]$ and $N$ is the number of problem instances for the task, and $s_i$ is the optional prompt defined as the instruction of the interpreter. The probability $p_{LLM}\left(r_i, y_j \mid \{x_i, r_i, y_i\}_{i \neq j}, x_j, s_i\right)$ generating the response $y_j$ is approximately invariant to the exact choice of the few-shot examples $S_j$.

**Algorithm.** Algorithm 1 summarizes the SuperContext augmentation method. The discriminative model $M$ is trained on the in-domain dataset $X$ and tested on the out-of-domain corpus $T$. For in-context learning of SuperContext, $y_j$ is prompted with the concatenation of the few-shot examples with the discriminative model's prediction, confidence, and the special prompt $s_i$. The output should be the prediction of LLMs towards the test case with interpretation if available.

---

**Algorithm 1** SuperContext for Natural Language Understanding

---

**Require:** In-domain Corpus $X$, Out-of-domain Corpus $Y$, A discriminative language model $M$, A large-scale generative model $L$, Instruction $R$, Output $O$,      ▷ The Instruction $R$ varies in per task.
**Ensure:** Predicted Labels for test cases in $Y$
1:   $M' \leftarrow \text{Finetune}(M, X)$
2:   **For** each test case $e_i$ in $Y$
3:     Confidence $c$, Predicted Label $l \leftarrow \text{Predict}(M', e_i)$
4:     $P \leftarrow \text{Concatenate}(R, e_i, l, c)$
5:     $O \leftarrow \text{Inference}(L, P)$
6:     **If** Interpretator Enabled **Then**
7:       **return** Interpretation, Predicted Label by Parser(O)
8:     **Else**
9:       **return** $O$

---

# 3 EXPERIMENTS

## 3.1 SETUP

**Source models.** As reported in GLUE-X (Yang et al., 2022), ELECTRA-large (Clark et al., 2020) achieves the best performance for both ID and OOD tasks over 21 small-scale pre-trained language models (maximum 774M parameters). Hence, we select ELECTRA-large as the SLM for NLU experiments, and RoBERTa-large (Liu et al., 2019) for QA. For evaluating the performance of SLM-enhanced LLMs, we select ChatGPT (OpenAI, 2023a) and Llama2-7B-chat (Touvron et al., 2023) as backbones, which are pre-trained on CommonCrawl, WebText, English Wiki, and others.

**Datasets.** We follow the OOD generalization setting of GLUE-X (Yang et al., 2022). In particular, we consider 7 classical NLU tasks: Sentiment Analysis (SA), Natural Language Inference (NLI), Paraphrasing, Question-Answering NLI (QNLI), Textual Entailment, Textual Similarity, and Linguistic Acceptability (Grammar). We sample 3,000 examples from GLUE-X for each OOD dataset and ensure that in-context samples are extracted from different domains of test sets. In total, Super-Context contains 43,728 instances on NLU for ChatGPT and 37,438 instances for Llama2-7B-chat.

Table 2: The table vividly displays the GLUE-X metrics garnered by diverse methods across 15 unique OOD datasets. 'AVG' denotes the average results across these 15 OOD datasets.

| Model | SST-2 OOD | MNLI OOD | QNLI OOD | RTE OOD | MRPC OOD | QQP OOD | STS-B OOD | CoLA OOD | Avg OOD |
|---|---|---|---|---|---|---|---|---|---|
| Human Performance | 97.69 | 91.80 | 92.33 | 91.12 | 83.50 | 79.13 | 92.62 | 66.47 | 86.83 |
| ELECTRA-large | 94.84 | 87.30 | 82.66 | 78.45 | 63.60 | 78.08 | 80.74 | 40.29 | 79.86 |
| ChatGPT | 94.83 | 41.54 | 81.82 | 68.56 | 60.23 | 43.23 | 72.61 | 39.05 | 66.67 |
| ChatGPT (+16-shot) | 94.72 | 64.24 | 74.14 | 68.34 | 60.91 | 74.24 | 64.60 | **47.15** | 72.28 |
| ChatGPT (+BM25) | 94.84 | 64.19 | 74.00 | 60.31 | **64.29** | 68.35 | 65.22 | 42.50 | 71.69 |
| SuperContext (w/o confidence) | 94.84 | 77.21 | 82.66 | 78.45 | 63.60 | 78.08 | **80.74** | 40.29 | 78.43 |
| SuperContext (+interpreter) | 94.84 | 80.73 | **83.81** | 78.60 | 64.26 | 77.80 | 76.15 | 39.47 | 78.77 |
| SuperContext (zero-shot) | **95.19** | **87.24** | 82.91 | **78.71** | 63.87 | **78.65** | 78.75 | 41.47 | **80.05** |
| ELECTRA-large | 95.42 | 87.29 | 82.69 | 78.84 | 37.59 | 77.18 | 80.74 | 45.73 | 76.84 |
| Llama2-chat | 90.56 | 34.30 | 66.85 | 60.77 | 36.20 | 51.57 | 37.12 | 6.94 | 55.92 |
| Llama2-chat (+16-shot) | 94.72 | 48.20 | 67.70 | 61.62 | 35.72 | 59.15 | 18.01 | 11.52 | 58.54 |
| Llama2-chat (+BM25) | 92.87 | 48.14 | 68.48 | 59.40 | 37.08 | 58.24 | 39.19 | 10.57 | 59.69 |
| SuperContext (zero-shot) | 94.95 | 85.45 | 81.60 | 78.39 | 36.70 | 61.79 | 45.67 | 40.84 | 73.89 |
| SuperContext (w/o confidence) | 94.29 | 76.68 | **82.66** | 78.46 | 43.41 | **78.17** | 80.74 | 40.26 | 75.68 |
| SuperContext (16-shot) | **95.45** | **87.14** | 82.17 | **79.07** | **54.63** | 77.18 | **80.74** | **45.47** | **79.08** |

**Baselines.** For NLU, we consider two in-context learning methods as baselines for ChatGPT (OpenAI, 2023a) and Llama2-7B-chat (Touvron et al., 2023), namely 16-shot ICL and BM25. The 16-shot ICL indicates the method that randomly extracts few-shot examples from the in-domain dataset as the demonstration prompt. While "+*BM25*" represents the dynamic in-context examples selection method using BM25 to select the top 16 examples that are similar to the test case. We also present the ablation that leverages SuperContext with the optional interpretation prompt, shown as "+*interpretor*". The variants of the backbone model are kept the same between ChatGPT and Llama2, namely "+*BM25*" and "+*16-shot*". Due to the relatively low instruction following ability of Llama2-7B-chat, the "+*interpretor*" is not explored in experiments of Llama2-7B-chat. Due to the difference in the instruction-following ability between the ChatGPT and Llama2-7B-chat, we insert the 16-shot in-context examples appended with the prediction and confidence of SLMs, namely SuperContext (16-shot). Human performance is extracted from GLUE (Wang et al., 2019).

**Evaluations.** Different from NLU, the question-answering task is evaluated by the hold-out test. The in-context examples are extracted from the training set and LLMs are evaluated on the validation set. We establish the baseline by using "*cluster+filter*" method. In particular, we adopt MiniLM (Wang et al., 2020) to encode the training examples and build a union-find set. Then, we use the cluster and filter pipeline to retrieve the most relevant examples with the test sample as in-context demonstrations for ChatGPT. For Llama2-7B-chat, we adopt two fine-tuned methods as baselines using multi-turn and single-turn tuning on 1.2 epochs, respectively. Notably, the total length of the prompt is controlled under 4,096, limited by Llama2.

## 3.2 NLU RESULTS

**Overall Performance.** The comprehensive results of natural language understanding tasks under the OOD evaluation are meticulously outlined in Table 2. Generally, SuperContext emerges as a dominant force, showcasing an elevated average result compared to both SLM (**80.05% vs. 79.86%**) and LLM (**80.05% vs. 66.67%**), underscoring the preeminent performance of SuperContext. Our experimental venture utilizing ELECTRA-large (334M Para.) to bolster Llama2-7B-chat's performance not only transcends ChatGPT (16-shot) (**79.08% vs. 72.28%**) but also parallels the SuperContext based on ChatGPT (79.08% vs. 80.05%), indicating its substantial capacity to markedly diminish inference costs. It is noteworthy that the data size used for ChatGPT and Llama2-7B-chat is different, leading to different results of SLMs (ELECTRA-large).

With the help of 16-shot in-context learning, the performance of ChatGPT can be improved from 66.67% to 72.28%, but still much lower than SuperContext (80.05% vs. 72.28%). The comparison between the in-context learning paradigm and our method proves that our method can outperform 16-shot in-context learning with a much shorter input sequence length (∼30 times).

Table 3: Results of ChatGPT and Llama2-7B-chat, and their variants on SQuAD 2.0. EM indicates the exact match and valid EM only accounts for the exact match of valid JSON. ACC No indicates the accuracy for no-answer questions and ACC accounts for the accuracy of has-answer questions.

| Model | Valid JSON | EM | Valid EM | ACC. No | ACC. Has. |
|---|---|---|---|---|---|
| SuperContext (zero-shot) | 85.18 | **57.68** | **57.81** | **54.65** | 60.71 |
| ChatGPT (cluster+filter) | 94.47 | 49.31 | 48.81 | 24.22 | 74.48 |
| ChatGPT (16-shot) | 99.49 | 44.69 | 44.52 | 13.22 | 76.25 |
| ChatGPT | 96.97 | 55.82 | 54.76 | 32.35 | **79.35** |
| SuperContext (16-shot) | 41.73 | **47.91** | 43.27 | **63.65** | 32.12 |
| Fine-tuned multi-turn | 96.40 | 25.70 | 26.66 | 10.47 | 40.16 |
| Fine-tuned single-turn | 97.17 | 47.22 | **48.60** | 39.44 | **55.02** |
| Llama2-7B-chat (16-shot) | 28.50 | 37.56 | 5.32 | 58.99 | 6.08 |
| Llama2-7B-chat | 40.09 | 46.48 | 40.13 | 3.72 | 31.87 |

We also present the results of SuperContext with the prompt of the interpreter, which requires LLM to recall influential in-context examples and output rationales when making the predictions, indicating as *SuperContext (+interpreter)*. To better understand the benefits of including the model confidence in the prompt, we present the results of SuperContext (w/o confidence). By comparing the results of SuperContext w/ and w/o confidence, we observe that including model confidence can bring significant improvements in the average performance for both ChatGPT and Llama2. Meanwhile, we find that for QNLI and QQP, Llama2 without the SLM's confidence achieves the best performance among several methods. Our results also indicate that the interpreter can not bring significant benefits when compared to SuperContext in most of the tasks, except a slight improvement can be achieved on QNLI. It can be because the explain-then-predict prompt (Wang et al., 2022a) may not be suitable for incorporating with SuperContext, leading to information overload.

**Llama2-7B-chat.** In addition to ChatGPT, we offer the comparison between SuperContext and several baselines based on the open-source model. Experimental results show that SuperContext with 16-shot in-context examples achieves the best results on seven of eight tasks included in GLUE-X compared to Llama2-7B-chat under the same setting without the help of the small model (**79.08% vs. 58.54%**). It is interesting to see that it outperforms ELECTRA-Large in terms of the average performance (**79.08 vs. 76.84**). Such a huge performance increase indicates that SuperContext improves the NLU capability of both Llama2-7B-chat and ELECTRA-large simply and effectively. In addition, we find that using BM-25 to retrieve the most relevant 16-shot examples of the test case is useful for improving the in-context learning performance (59.69% vs. 58.54%).

**Task-level Analysis.** On the task level, we observe that both ChatGPT and Llama2 show a relatively lower accuracy than the expectation on multiple tasks, including OOD evaluation on MNLI, MRPC, and QQP. For example, the original ChatGPT and Llama2-7B-chat can only achieve 41.54% and 34.30% on MNLI, respectively. With the help of SuperContext, MNLI-OOD results can be improved to 87.24% and 87.14% on ChatGPT and Llama2-chat, respectively. For STS-B which is a textual similarity task, we find that the original Llama2-chat model performs poorly with or without in-context learning and the zero-shot performance of Llama-2-chat is significantly lower than ChatGPT (37.12% vs. 72.61%). Notably, although the zero-shot performance of SuperContext based on Llama2-7B-chat is lower than ChatGPT using the same setting on all tasks, SuperContext based on 16-shot Llama2-7B-chat can even beat SuperContext based on zero-shot ChatGPT in multiple OOD tasks, including SST-2, RTE, STS-B, and CoLA, representing the efficacy of our method not only for proprietary LLMs but also for relatively small-scale models, Llama2-7B-chat.

### 3.3 QA RESULTS

The fact-conflicting of LLMs is considered a core issue in LLMs because it is challenging for users to be aware of and may pose misinformation dissemination. We evaluate LLMs' ability towards minimizing the hallucination on the QA task based on SQuAD 2.0 (Rajpurkar et al., 2018), which is a suitable testbed since it can be addressed using both discriminative and generative manners.

**Results of ChatGPT.** The results are presented in Table 3. We find that although the original ChatGPT can achieve the highest accuracy for deterministic questions (79.35%), the exact match (EM) and accuracy for open questions can be significantly improved by SuperContext. In particular, the accuracy for no_answer questions can be improved from **32.35%** (ChatGPT) to **54.65%** (SuperCon-

text), indicating the huge benefits. Besides, we find that even with the careful design of in-context learning prompts and filter methods, SuperContext still outperforms two in-context learning variants in terms of all metrics, indicating that pure in-context learning without fine-tuning LLMs brings no benefit to the QA task. Furthermore, SuperContext even outperforms the fine-tuned method in a multi-turn setting on all metrics. We believe that such a huge performance benefit (54.65% vs. 13.22%) compared to the traditional 16-shot in-context method when answering no_answer questions ("ACC.NO.") proves that results achieved by discriminative models are effective enough to reduce the hallucination.

**Results of Llama2-7B-chat.** We observe that the fine-tuned methods can significantly improve the rate of valid JSON. In particular, the fine-tuned single-turn method improves the valid JSON of the original Llama2-chat from 40.09% to 97.17% and achieves the best performance for valid EM (48.6%) and accuracy for has-answer questions (55.02%). Despite fine-tuned methods outperforming the original Llama2-chat and the in-context learning version, SuperContext achieves the best performance in terms of the EM and accuracy for no-answer questions. We observe that the original Llama2-7B-chat model struggled with format adherence and hallucinations, especially in answering no-answer questions. This is reflected in the notably low score of 3.72. In other words, it cannot output "I don't know" when the question is unanswerable. However, when applying in-context learning with a mix of no-answer and has-answer instances, we noticed an improvement in handling no-answer questions, though this came at the cost of reduced accuracy in has-answer questions.

## 4 ANALYSIS AND DISCUSSION

### 4.1 REVERSED PREDICTIONS

As displayed in Table 4, we study the difference between the final prediction of LLMs and the prediction of SLMs. The detailed task-level performance is shown in the Appendix. The results demonstrate that predictions of 3.02% instances have been overridden during the inference face of ChatGPT by using SuperContext. 57.88% of them have been corrected, indicating that the reference generated by SLMs brings

Table 4: Statistics of reversed predictions. "%Reversed" denotes the percentage of LLMs' predictions that differ from the predictions of SLMs. "Reversed Acc." is short for the possibility of the reversed predictions that from incorrect to correct.

| Method | %Reversed | Reversed Acc. |
|---|---|---|
| SuperContext (ChatGPT) | 3.02% | 57.88% |
| SuperContext (Llama2-7B-chat) | 0.50% | 52.13% |

positive benefits for improving the NLU capability of LLMs. SuperContext on Llama2-7B-chat exhibits a relatively lower possibility of reversing the prediction of SLMs (0.5%), yet also inspires LLMs to correct SLMs' predictions in a more accurate direction than the random guess (52.13%).

### 4.2 INTERPRETATION ANALYSIS

In addition to the prediction results, we are also interested in understanding the reason behind the result that SuperContext significantly outperforms the traditional in-context learning method. We aim to answer this question from two aspects, how LLMs recall already learned concepts and rationale from pre-training (Han et al., 2023; Gu et al., 2023) and why it fails in the OOD setting.

**Learning from In-context Demonstrations.** We explore how language models use long contexts. Figure 3 shows the influence of demonstrations during the inference stage, where the y-axis indicates how many times ChatGPT and InstructGPT take the $i_{th}$ in-context example as the emphasized one towards the prediction. The x-axis is sorted by the order of occurrence of in-context examples over 8 natural language understanding tasks. As shown in the figure, both ChatGPT and InstructGPT show a significant occurrence times difference among in-context examples. In particular, ChatGPT with 16-shot examples shows a trend of decreasing attention with the order of appearance. For example, the second in-context example has been paid attention to over 35,000 times while the last example only receives around 5,000 times attention. In terms of InstructGPT, we observe distinctive U-shaped occurrence times, which can be visualized in Figure 3(b). We find that the model tends to pay attention to the beginning or the end of the input context (in-context examples), and the attention significantly degrades in the middle of long contexts. This observation is consistent with the findings of (Liu et al., 2023) on the use of long contexts when performing downstream tasks, which suggests that model performance significantly degrades when models must access relevant

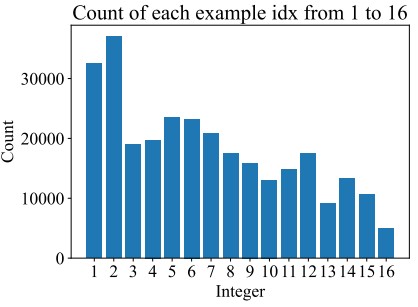

(a) Interpretation results of ChatGPT.

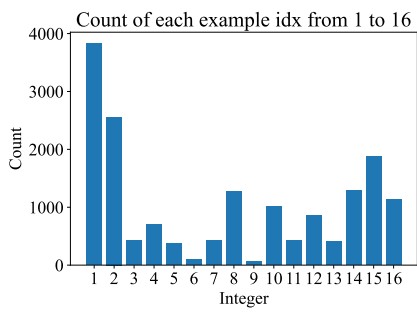

(b) Interpretation results of InstructGPT.

Figure 3: Counting the times of 16-shot in-context examples that have been considered as the influential examples over 8 NLU tasks, sorting by order of occurrence.

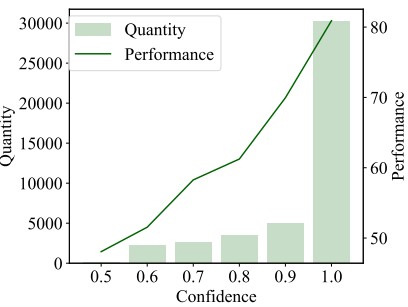

(a) The calibration laws of ChatGPT.

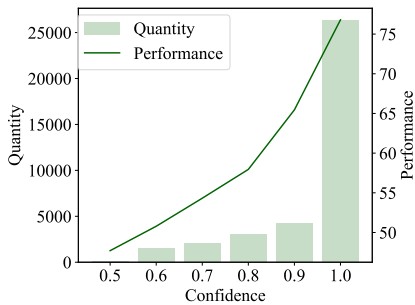

(b) The calibration laws of Llama2-7B-chat.

Figure 4: The correlation between the SLM confidence and LLM performance evaluated on the GLUE-X benchmark. The dark green line represents the normalized performance of LLMs using SuperContext corresponding with the right y-axis while the light green bar indicates the volume of instances with the specific confidence interval corresponding with the left y-axis.

information in the middle of long contexts and provide a new perspective for future long-context models. We also collect the sentence-level rationale generated by LLMs when making predictions, and count for the word frequency for each task of GLUE-X based on ChatGPT, aiming to provide the internal causes of OOD generalizability. However, the rationale is generated by LLMs and thus may contain hallucinations, which should be treated with caution and just for reference.

## 4.3 THE EFFECT OF SLM CONFIDENCE

Since we rely on the complementarity between SLMs and PLMs, SLMs must convey its *certainly* in task knowledge and *uncertainly* in domain knowledge to PLMs. The confidence score in the design serves a crucial role in such communication channels. We show the correlation between the confidence of SLMs and the prediction performance of LLMs. As shown in Figure 4, both ChatGPT and Llama2-7B-chat demonstrate a positive correlation between SLMs' confidence and LLM' performance, representing a high consistency between those models. The x-axis represents the confidence interval covering from 0.4-1.0, for example, 0.5 indicates the instances with the prediction confidence between 0.4-0.5. It is noteworthy that the confidence is computed by the zero-shot test based on SLMs trained on unseen domains, which indicates that high confidence requires a decent generalization ability of small models. We speculate that SuperContext shows superior performance than both SLMs and LLMs since it leverages the benefits of high consistency in discriminative models and the complementarity property of recent generative models. Besides, such a positive calibration law underscores the importance of involving both prediction and confidence in the prompt design of SuperContext. The data statistic of data quantity shows that most instances included in GLUE-X receive the highest confidence interval from 0.9 to 1.0, and this part of the data can be predicted with significantly higher accuracy than others. By comparing the experimental results of GPT-3.5 and Llama2-7B-chat, we find that when the confidence is more than 0.6, the average performance of GPT-3.5 is substantially better than Llama2-7B-chat.

## 5    RELATED WORK

**In-context Learning.**   Scaling up pre-trained language models stimulates the in-context learning ability is first introduced by GPT-3 (Brown et al., 2020), introducing the potential to accurately comprehend instructions and complete complex tasks with no supervision (Chowdhery et al., 2022; OpenAI, 2023b; Sun et al., 2023a). As evidenced by previous work (Shwartz et al., 2020; Nye et al., 2021; Perez et al., 2021), the ICL performance can be significantly enhanced by incorporating auxiliary knowledge or reasoning instructions in a prompt, such as Chain-of-Thought (COT) (Wei et al., 2022) and Tree-of-Thoughts (TOT) (Yao et al., 2023). However, such a multi-step reasoning process could be tedious and expensive to use (assuming we perform ICL for GPT-4), whereas our method is cost-efficient since the supervised knowledge occupies only a short length in the prompt.

There is a line of work for improving the in-context learning performance by either constructing demonstrations (Arora et al., 2022; Si et al., 2022; Lyu et al., 2022; Gu et al., 2023; Ye et al., 2023; Dhuliawala et al., 2023) or framing an exploration of example selection methods (Wu et al., 2023; Wang et al., 2023b; Sun et al., 2023a; Agrawal et al., 2022; Wang et al., 2022c;b; Lu et al., 2022; Wang et al., 2023c) and even order (Lu et al., 2021; Zhao et al., 2021; Liu et al., 2021; 2023). The contrastive decoding method (Li et al., 2023b) considers the assistance smaller language model but requires external computation. Differently, SuperContext demonstrates its superior performance on OOD test data in a cost-effective manner.

Our work is also connected with work focusing on understanding and explaining in-context learning from different perspectives, including the implicit Bayesian Inference (Xie et al., 2021), pre-training data (Han et al., 2023; Pan et al., 2023), and information compression (Wang et al., 2023a; Wu et al., 2023). Different ways of understanding ICL in realistic NLP tasks have been proposed before (Min et al., 2022; Dong et al., 2022; Wang et al., 2023b), the interpretation part in SuperContext aims to answer how LLMs recall in-context examples and output rationale.

**Knowledge in Context.**   Using external knowledge as auxiliary information to assist LLMs in providing truthful and timely responses represents an emerging solution (Mialon et al., 2023; Xiao et al., 2023) in recent. Traditional retrieve-based methods (Rubin et al., 2021; Ni et al., 2021; King & Flanigan, 2023) require a knowledge retriever as the prior step for guiding the generation of responses. Besides, the external knowledge source could extend beyond local documents to encompass the entire Internet (Ni et al., 2021; Gao et al., 2023). In addition, LLMs can leverage special plug-ins to improve their capabilities, such as Toolformer (Schick et al., 2023) and LangChain (Chase, 2022) for calling external APIs, and HuggingGPT (Shen et al., 2023) for using models.

Previous work either relies on web information and search engines for gaining external knowledge (Yu et al., 2023) or accomplishes planning tasks outside the NLP scope. (Xu et al., 2023a) evaluates the efficacy of small language models as plug-ins under an in-domain setting using GLUE and lacks an interpretation part to explain the reasons. SuperContext shares a conceptual similarity with SuperICL (Xu et al., 2023a) and HuggingGPT (Shen et al., 2023) in leveraging language model architectures. However, the key distinction lies in our approach's application and analysis under out-of-distribution (OOD) conditions, a less explored area in the existing literature.

## 6    CONCLUSION AND FUTURE WORK

We constructed SuperContext, an SLM-LLM interaction framework using supervised knowledge for making LLMs better in-context learners in the OOD natural language understanding benchmark and text generation settings. Our goal is to improve the generalizability and factuality of LLMs using cost-efficient, task-specific, and generalizable SLMs. Results on 8 NLU tasks and 1 generation task show that (1) current in-context learning methods still lag much behind humans towards the OOD evaluation of NLU and hold-out test of QA; (2) the traditional in-context learning paradigm faces the forgetting problem and is limited by the input sequence length; (3) SuperContext can bring decent performance benefit compared to few-shot in-context learning and outperform original SLMs and LLMs with both zero-shot and few-shot settings. In the future, we anticipate expanding the scope of SuperContext to cover additional text generation tasks and exploring its effectiveness in various real-world applications.

## ACKNOWLEDGEMENT

We would like to thank the anonymous reviewers for their insightful comments and suggestions to help improve the paper. This publication has emanated from research conducted with the financial support of the Pioneer and "Leading Goose" R&D Program of Zhejiang under Grant Number 2022SDXHDX0003 and the National Natural Science Foundation of China Key Program under Grant Number 62336006.

## ETHICAL STATEMEMT

**Ethical Use of ChatGPT and InstructGPT.** In adherence to the official guidelines provided by OpenAI, we utilized ChatGPT (gpt-3.5-turbo) and InstructGPT (text-davinci-003), setting the temperature of all tasks to zero to ensure reproducibility. For experiments conducted on the SQuAD 2.0 dataset, we employed gpt-3.5-turbo-16k to ensure the prompt length remained within the model's window length.

**Social Impact.** The primary objective of this study is to repurpose the extensively labeled data in specific domains, which required substantial human and material resources to generate. We aim to use these data to train a task-specific model to assist LLMs in mitigating hallucinations produced during Natural Language Understanding (NLU) and Question Answering (QA) tasks, thereby enhancing the safety of the LLMs. Notably, all of datasets involved in this work belong to the publicly available detasets, and thus do not contain any personal privacy data.

**Potential Concerns.** We acknowledge several limitations of this study and propose a series of open questions for subsequent research. We discuss the potential concerns and limitations of this work.

1. Exploration of Other Large-Scale Language Models: In this study, we delve into the examination of ChatGPT and Llama2. Nevertheless, a plethora of recently proposed models, such as GPT-4, PaLM, Falcon, and Claude, beckons for comprehensive analysis. This work does not involve any commercial competition and belongs to non-profit research.

2. Unveiling More Properties of LLMs: This work investigates the generalizability and factuality of LLMs, yet uncharted territories remain. The exploration of social bias and the reasoning capacity of LLMs promises to be an interesting avenue for further research. We respect the human rights of all people and ensure that crowdsourcing workers are adequately paid for this work.

3. In-Depth Analysis for In-Context Learning Understanding: SuperContext relies on the complementarity between SLMs and PLMs, where SLMs must convey its *certainly* in task knowledge and *uncertainly* in domain knowledge to PLMs. A pivotal question persists: **can this complementary behavior be attributed to the pre-training data or a handful of few-shot demonstrations?** We plan to refine the interaction mechanism between SLM and LLM to further understand in-context learning. Our current analysis does not involve any personal privacy.

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

## A    ADDITIONAL RESULTS: GENREALIZABILITY

We provide full experimental results in Table 5. Different from Table 2, which demonstrates the average result of each task included in GLUE-X, we present a fine-grained analysis to show the efficacy of our method and differences among tasks. In particular, it is interesting to see that with the help of SuperContext, both ChatGPT and Llama2-7B-chat surpass the supervised task-specific model, ELECTRA, in terms of higher average performance, indicating that SuperContext introduces the benefits of complementarity to enhance the generalizability of LLMs' towards the NLU tasks. Such a task-level analysis also sheds light on future work to design task-specific methods.

Table 5: Performance evaluation of SuperContext and several baselines based on GLUE-X dataset. The table showcases the detailed evaluation results of SLMs, LLMs, and SuperContext. C. is short for ChatGPT and L. represents Llama2-7B-chat.

| ID | OOD | Ours (C.) | ELECTRA (C.) | ChatGPT (16) | ChatGPT | ELECTRA (L.) | Llama2 | Ours (L.) |
|---|---|---|---|---|---|---|---|---|
| SST2 | IMDB | 93.97 | 94.87 | 94.03 | 93.63 | 94.97 | 91.48 | 94.97 |
| SST2 | YELP | 97.30 | 96.63 | 97.00 | 96.87 | 97.39 | 95.63 | 97.53 |
| SST2 | Amazon | 95.87 | 95.37 | 94.99 | 94.33 | 95.84 | 94.76 | 95.84 |
| SST2 | Flipkart | 93.60 | 93.50 | 92.84 | 94.50 | 93.49 | 89.59 | 93.45 |
| CoLA | Grammar | 41.47 | 40.29 | 47.15 | 39.05 | 45.73 | 11.52 | 45.47 |
| MRPC | QQP | 55.06 | 54.36 | 66.77 | 69.94 | 27.66 | 32.97 | 57.22 |
| MRPC | Twitter | 72.68 | 72.83 | 55.05 | 50.52 | 47.52 | 38.46 | 52.04 |
| QQP | MRPC | 80.17 | 79.56 | 79.49 | 42.04 | 80.74 | 17.99 | 80.74 |
| QQP | Twitter | 77.13 | 76.59 | 68.99 | 44.41 | 78.94 | 70.35 | 78.94 |
| MNLI | MNLI_mis | 88.67 | 89.13 | 64.77 | 43.85 | 75.42 | 47.94 | 75.42 |
| MNLI | SNLI | 85.80 | 85.47 | 63.70 | 39.23 | 89.13 | 56.30 | 89.10 |
| RTE | HANS | 72.87 | 72.87 | 58.20 | 56.73 | 85.45 | 40.10 | 85.18 |
| RTE | SCITAIL | 84.55 | 84.02 | 62.54 | 80.38 | 73.68 | 60.07 | 74.34 |
| QNLI | NewsQA | 82.91 | 82.66 | 74.14 | 81.82 | 84.00 | 63.17 | 83.79 |
| STS-B | SICK | 78.75 | 80.74 | 64.60 | 72.61 | 82.69 | 67.69 | 82.17 |
| GLUE-X | AVG. | **80.05** | 79.86 | 72.28 | 66.67 | 76.84 | 58.53 | **79.08** |

## B    TASK-LEVEL STATISTICS OF REVERSED PREDICTIONS

We present the detailed analysis of predictions of ELECTRA-large reversed by ChatGPT and Llama2-7B-Chat, along with the number of test instances for each task in Table 6. In general, we observe that ChatGPT demonstrates a superior ability to reverse predictions of ELECTRA-Large compared to Llama2-7B-chat, aiming to correct errors when making OOD predictions on NLU tasks. On the other hand, ChatGPT exhibits higher accuracy in making modifications that override classification results compared to Llama2-7B-chat.

The OOD testing on QNLI is perceived by both models to contain the highest proportion of data that should have the final decision overridden. Specifically, 15.11% of the test data is amended by ChatGPT, while 2.49% of the data is reversed by Llama2-7B during the inference stage. Naturally, for tasks with relatively lower error rates, such as SST-2 and MNLI, the probability of the models making modifications is also low. This underscores the significant ability of larger models to evaluate the predictions and confidence levels of task-specific fine-tuned models.

In terms of the Reversed Accuracy (where the probability of random guess is 50%) as shown in Table 6, we find that ChatGPT exhibits a higher correction accuracy than random guessing on seven out of eight tasks. In contrast, Llama2-7B-chat is capable of reversing predictions in only six out of the eight tasks and surpasses random guesses in only half of the tasks.

## C    ADDITIONAL RESULTS OF CALIBRATION LAWS

We supplement the results of the calibration laws with two additional model groups, namely ELECTRA-large and InstrutGPT, in Figure 5. Consistent with ChatGPT and Llama2-7B-chat,

Table 6: The detailed statistics of reversed predictions on each task of GLUE-X. "%Reversed" denotes the percentage of predictions of LLMs that differ from the predictions of SLMs. "Reversed Acc." is short for the possibility of the reversed predictions that from incorrect to correct. "%Error" is the error rate of the ELECTRA-large baseline. "#Instances" is the total number of test samples.

| Model | Metric | SST2 | MNLI | QNLI | RTE | MRPC | QQP | STS-B | CoLA |
|---|---|---|---|---|---|---|---|---|---|
| ChatGPT | %Error | 5.16 | 12.70 | 17.34 | 21.55 | 36.40 | 21.92 | 19.26 | 59.71 |
| | %Reversed | 0.81 | 1.10 | 15.11 | 3.92 | 1.40 | 1.92 | 8.27 | 2.87 |
| | Reversed Acc. | 71.13 | 42.42 | 50.81 | 53.93 | 82.14 | 68.70 | 53.23 | 74.42 |
| | #Instances | 12,000 | 6,000 | 2,866 | 4,862 | 6,000 | 6,000 | 3,000 | 3,000 |
| Llama2-7B-Chat | %Error | 4.58 | 12.71 | 17.31 | 21.16 | 62.41 | 22.82 | 19.26 | 54.27 |
| | %Reversed | 0.04 | 0.28 | 2.49 | 0.87 | 2.05 | 0 | 0 | 0.25 |
| | Reversed Acc. | 80.00 | 23.53 | 39.44 | 65.85 | 88.98 | 0 | 0 | 16.77 |
| | #Instances | 10,549 | 5,996 | 2,849 | 4,738 | 2,604 | 5,326 | 3,000 | 2,376 |

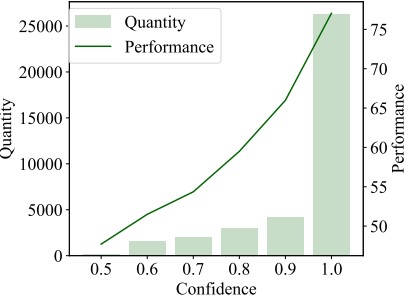

(a) The calibration laws of ELECTRA-large.

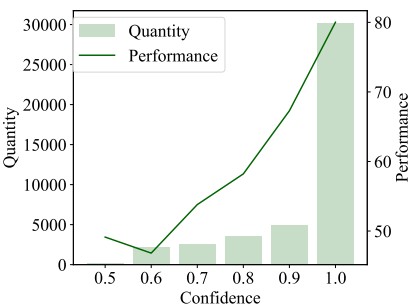

(b) The calibration laws of InstructGPT.

Figure 5: The calibration laws of ELECTRA-large and InstructGPT between the confidence and performance evaluated on the GLUE-X benchmark. The dark green line represents the LLMs' performance using SuperContext corresponding with the right y-axis while the light green bar indicates the volume of instances with the specific confidence interval corresponding with the left y-axis.

both ELECTRA-large and InstructGPT exhibit a positive correlation between confidence and performance. Distinctively, the curve for InstrutGPT demonstrates more pronounced fluctuations, especially when the confidence is relatively low.

## D  DETAILED PROMPT DESIGNS

We present the detailed prompt designs for each task, using SST-2, CoLA, and SQuAD2.0 as demos.

### D.1  NATURAL LANGUAGE UNDERSTANDING

**SST-2**

```
SST-2: You are tasked with predicting the sentiment of a given sentence
as either 'positive' or 'negative'. Use the prediction from the
pre-trained model (334M Parameters) fine-tuned on a sentiment analysis
dataset as a reference to aid your judgment.

Test Case: Sentence: "[Input]" Model's Prediction: Model's Confidence:

Please provide your analysis using the format below and then give your
final prediction:

1. Influence Degree: On a scale of 0 to 1 (in increments of 0.1), how
much did the fine-tuned model's prediction influence your judgment? 2.
Critical Features: Identify the specific word or phrase in the test case
that played a pivotal role in your prediction of sentiment analysis.

After analyzing, please provide your final prediction.
```

**CoLA**

CoLA: You are tasked with predicting the sentiment of a sentence's grammar as either 'acceptable' or 'unacceptable'. Use the prediction from the pre-trained model (334M Parameters) fine-tuned on a grammar test dataset as a reference to aid your judgment.

+ Test case: \[Test case]" Model's Prediction: Model's Confidence:

Please provide your analysis using the format below and then give your final prediction:

1. Influence Degree: On a scale of 0 to 1 (in increments of 0.1), how much did the fine-tuned model's prediction influence your judgment?

2. Critical Features: Identify the specific word or phrase in the test case that played a pivotal role in your grammar-acceptable prediction.

After analyzing, please provide your final prediction.

## D.2 QUESTION ANSWERING

**SQuAD 2.0**

SQuAD 2.0: [INST] <<SYS>> You are a helpful, respectful and honest assistant. Always answer as helpfully as possible, while being safe. Your answers should not include any harmful, unethical, racist, sexist, toxic, dangerous, or illegal content. Please ensure that your responses are socially unbiased and positive in nature. If a question does not make any sense, or is not factually coherent, explain why instead of answering something not correct. If you dont́ know the answer to a question, please dont́ share false information. <</SYS>>

Extract from the following context the minimal span word for word that best answers the question. Think step by step and explain your reasoning. Then give the answer in JSON format as follows: ```json "answer": ... ``` If the answer is not in the context, the answer should be exactly a string "?", this is very important. Context: context Question: question Hereś a potential answer to the question: ```json "answer": ["answer"] ```

