# OpenReview forum: "Supervised Knowledge Makes Large Language Models Better In-context Learners"
_ICLR.cc/2024/Conference — ICLR 2024 poster_

### Official Review · Reviewer_ucxK · 2023-10-30

**Soundness:** 3 good
**Presentation:** 3 good
**Contribution:** 3 good
**Rating:** 5
**Confidence:** 3

**Summary:**

This paper presents an SLM-LLM integration framework, which aims to inject supervised knowledge into LLMs with the aid of the task-specific language model. The motivation behind is that SLMs have more task knowledge thanks to the supervised training while LLMs capture more general knowledge with large-scale pretraining. The methodology is simple and straightforward, i.e., directly incorporating the predictive results from SLMs with the prompts. Experiments are conducted on a set of natural language understanding tasks and a QA task. Some performance improvements are observed under their OOD settings.

**Strengths:**

1. It is a good motivation to enhance LLMs' task-specific ability with the aid of supervised models trained on the task.

2. Some performance improvements are observed.

3. The writing of the paper is overall good.

**Weaknesses:**

1. The novelty of the method is not significant. The technical contribution of the paper is insignificant.

2. The tested tasks, i.e., NLU and QA, are too simple to illustrate the authors' statements. I would like to see more positive results on more challenging tasks, such as reasoning or code generation.

3. I have some questions regarding the experiments. See the question part.

**Questions:**

Q1. The supervised model is trained on the in-domain corpus and then used to enhance the inference of LLMs. Have you tried fine-tuning the LLMs with the in-domain corpus? Although it has a much higher cost, I wonder whether there is a much larger performance improvement after SFT.

Q2. I cannot capture the motivation of presenting Section 4.2. It seems that this section is not related to your core idea of the paper.

Q3. Apart from appending the model prediction to the prompt, you also include the model confidence score to the prompt. How much does the framework benefit from this design? If removing the model confidence score, how does the performance change?

Q4. How do you explain the extremely low scores in Table 3, i.e., 5.32, 6.08 and 3.72?

---

> ### Author Response · Authors · 2023-11-22
> **Response to Reviewer ucxK**
>
> We sincerely thank you for your insightful comments and valuable suggestions on our manuscript. Your expertise and detailed review have been instrumental in enhancing the quality and clarity of our work. For your convenience, we have marked all the related revisions in blue within the manuscript, ensuring easy identification of the changes made. We welcome any further inquiries or suggestions you might have, as your input is invaluable in refining our research.
>
> ### Question 1:
> The supervised model is trained on the in-domain corpus and then used to enhance the inference of LLMs. Have you tried fine-tuning the LLMs with the in-domain corpus? Although it has a much higher cost, I wonder whether there is a much larger performance improvement after SFT.
>
> ### Answer 1:
> We appreciate your suggestion regarding instruction-tuning the LLMs with the in-domain corpus. Existing literature indicates that while tuned LLMs demonstrate high performance on many reasoning datasets, they still face challenges with out-of-distribution tests. This insight, derived from recent studies such as Liu et al. (2023), informed our decision to focus on enhancing LLM inference with supervised models trained on in-domain data, a less explored approach with potential benefits in handling OOD scenarios.
>
> [1] Liu, Hanmeng, et al. "Evaluating the logical reasoning ability of chatgpt and gpt-4." arXiv preprint arXiv:2304.03439 (2023).
>
> ### Question 2:
> I cannot capture the motivation of presenting Section 4.2. It seems that this section is not related to your core idea of the paper.
>
> ### Answer 2:
> Thank you for highlighting the need for clarity regarding Section 4.2. The purpose of this section is to investigate the limitations of in-context learning, as observed in Table 2, where LLMs underperform in this setting compared to our SuperContext approach with zero-shot demonstrations. We designed two interpretation tasks in Section 4.2 to delve into why in-context learning might be less effective, thereby providing a deeper understanding of the dynamics between LLMs and supervised learning models. This analysis is crucial for comprehending the full scope of our proposed methodology.
>
> ### Question 3:
> Apart from appending the model prediction to the prompt, you also include the model confidence score to the prompt. How much does the framework benefit from this design? If removing the model confidence score, how does the performance change?
>
> ### Answer 3:
>
> | **Model (ChatGPT)** | **SST-2** | **MNLI** | **QNLI** | **RTE** | **MRPC** | **QQP** | **STS-B** | **CoLA** | **Avg** |
> |----------|----------|----------|----------|----------|----------|----------|----------|----------|----------|
> SuperContext (w/o confidence) | 94.84 | 77.21 | 82.66 | 78.45 | 63.60 | 78.08 | **80.74** | 40.29 | 78.43
> SuperContext (zero-shot) | **95.19** | **87.24** | **82.91** | **78.71** | **63.87** | **78.65** | 78.75 | **41.47** | **80.05**
>
> | **Model (LLaMA2-7B)** | **SST-2** | **MNLI** | **QNLI** | **RTE** | **MRPC** | **QQP** | **STS-B** | **CoLA** | **Avg** |
> |----------|----------|----------|----------|----------|----------|----------|----------|----------|----------|
> SuperContext (w/o confidence 16-shot) | 94.29 | 76.68 | 82.66 | 78.46 | 43.41 | **78.17** | **80.74** | 40.26 | 75.68
> SuperContext (16-shot) | **94.95** | **87.14** | **82.17** | **79.07** | **54.63** | 77.18 | **80.74** | **45.47** | **79.08**
>
> Your inquiry into the role of model confidence in our framework is useful. We have incorporated ablation studies in Table 2 to address this. The results show that including the confidence score of small models significantly boosts the performance of LLMs in various NLU tasks. For instance, the performance of ChatGPT improves from an average of 78.43 to 80.05 when confidence scores are included. Similarly, LLaMA2-7B's performance increases from 75.68 to 79.08, demonstrating the effectiveness of integrating model confidence in our approach.
>
> ### Question 4:
> How do you explain the extremely low scores in Table 3, i.e., 5.32, 6.08 and 3.72?
>
> ### Answer 4:
> The low scores in Table 3 primarily result from the models' challenges in generating correctly formatted responses and handling no-answer scenarios. We observed that the original Llama2-7B-chat model struggled with format adherence and hallucinations, especially in answering no-answer questions. This is reflected in the notably low score of 3.72. In other words, it cannot output ``I don't know`` when the question is unanswerable. However, when applying in-context learning with a mix of no-answer and has-answer instances, we noticed an improvement in handling no-answer questions, though this came at the cost of reduced accuracy in has-answer questions.  These observations and their implications are further discussed in the revised manuscript following Table 3.

---

> > ### Author Response · Authors · 2023-11-23
> >
> > We welcome any further inquiries or suggestions you might have, as your input is invaluable in refining our research. Thank you for your continued guidance and support in this process.

---

### Official Review · Reviewer_9mey · 2023-11-01

**Soundness:** 2 fair
**Presentation:** 1 poor
**Contribution:** 3 good
**Rating:** 5
**Confidence:** 3

**Summary:**

This paper proposes an innovative in-context learning method through integrating the outputs from small discriminative models fine-tuned on supervised knowledge into LLM prompts. The outputs from small discriminative models are expected to include the importance prediction of different in-context examples for the current test case, and the corresponding explanation for such importance prediction. Experiments demonstrate that the proposed method effectively enhance the performance of LLM on four tasks.

**Strengths:**

1. This paper provides a pioneering approach to systematically integrate supervisedly fine-tuned models into LLM inference.

2. The proposed method significantly improves the performance of LLM, especially in managing OOD data and mitigating hallucinations.\

3. Compared with recent work Li et al. (2023b), the proposed approach is cheap and generalized.

**Weaknesses:**

1. Illustration of the proposed method in Figure 1 is not clear. It is unclear how the supervised knowledge participate in the whole procedure, while the capture states that it plays a core role.

2. Equation (1) is misleading. On one hand, the left term contains "\{x_i,y_i\}_{i!=j}" as part of the condition for LLM inference, while "i!=j" is not a complete expression for the range of "i". On the other hand, the right term contains "\{x_i,y_i\}_{i \in S_j \subset [1,...,N]", i.e., the few-shot examples, as part of the condition, which may conflict with the statement in context: "where i \in [1,...,N]". I can only assume the author intended to mean "\{x_i,y_i\}_{i!=j, i \in [1,...,N]}" in the left term instead of the right term. However, the following context called this condition as "the concatenation of the task description", which leads to puzzle again.

3. The explanation of proposed method in section 2.2 is not clear enough or may have grammar error:
a. "Specifically, our receipt r_i": receipt from what? in form of what?
b. "learning from which ... and which ... is important": who learn what? how does it learn?
c. "as typical data points are OOD for the model": for the discriminative model or for the LLM?

4. Table 2 has two lines for the same baseline "ELECTRA-large" with different scores and no explanation for such difference. Specifically, the scores of ELECTRA-large in those two lines have small divergence except that it got 63.60 on MRPC in the first line but 37.59 in the second line. This raises serious questions about the reliability of the experimental results.

5. The improvement of proposed method in managing OOD data is not obvious or even negative. For example, equipped with both ELECTRA-large and LLMs, the scores of proposed method only surpass ELECTRA-large within 1 point on most datasets in Table 2, and even become less than ELECTRA-large on some of the datasets. Nevertheless, there is no analysis for the casualty in evaluation, e.g., the variance of those scores.

**Questions:**

1. How is the discriminative model trained?

2. The discriminative model provides the index of influential in-context examples, so do you rerank all the in-context examples according to this signal before/during concatenation?

---

> ### Author Response · Authors · 2023-11-22
> **Response to Reviewer 9mey (1/2)**
>
> We sincerely thank you for your insightful comments and valuable suggestions on our manuscript. Your expertise and detailed review have been instrumental in enhancing the quality and clarity of our work. In response to your comments, we have carefully examined and addressed each issue you raised.
>
> For your convenience, we have marked all the related revisions in blue within the manuscript, ensuring easy identification of the changes made. We welcome any further inquiries or suggestions you might have, as your input is invaluable in refining our research.
>
> The most important concern about the reliability of the experimental results has been explained in Question 4.
>
> ### Question 1:
> The illustration of the proposed method in Figure 1 is not clear. It is unclear how the supervised knowledge participate in the whole procedure, while the capture states that it plays a core role.
>
> ### Answer 1:
> Thank you for your feedback. We acknowledge the need for clearer representation of supervised knowledge in our methodology. Accordingly, we have revised the caption of Figure 1 to better illustrate how supervised knowledge plays a pivotal role in our approach, clarifying its integration and significance in the overall process.
>
> ### Question 2:
>
> Equation (1) is misleading. On one hand, the left term contains "{x_i,y_i}{i!=j}" as part of the condition for LLM inference, while "i!=j" is not a complete expression for the range of "i". On the other hand, the right term contains "{x_i,y_i}{i \in S_j \subset [1,...,N]", i.e., the few-shot examples, as part of the condition, which may conflict with the statement in context: "where i \in [1,...,N]". I can only assume the author intended to mean "{x_i,y_i}_{i!=j, i \in [1,...,N]}" in the left term instead of the right term. However, the following context called this condition "the concatenation of the task description", which leads to a puzzle again.
>
> ### Answer 2:
>
> We appreciate your insightful observations regarding Equation 1. You are correct in identifying a discrepancy. The intended meaning is that the term {x_i,y_i}_{i!=j, i \in [1,...,N]} applies to the left side of the equation. We have now rectified this in the manuscript to accurately reflect the experimental setup and avoid any confusion. The revised text should now provide a clear and accurate depiction of our method.
>
> ### Question 3:
> a. "Specifically, our receipt r_i": receipt from what? in the form of what?
> b. "learning from which ... and which ... is important": who learn what? how does it learn?
> c. "as typical data points are OOD for the model": for the discriminative model or for the LLM?
>
> ### Answer 3:
>
> Thank you for highlighting these ambiguities. We have made the following revisions for clarity:
>
> Question A: The term 'receipt r_i' refers to the output and confidence score generated by Supervised Language Models (SLMs), in form of the prompt.
> Question B: We have elaborated on how Large Language Models (LLMs) are trained to learn from in-context examples, providing detailed descriptions of the prompt design and learning process.
> Question C: The term 'Out-Of-Distribution (OOD)' applies to both discriminative models and LLMs in our context. The training set and in-context examples are both drawn from the in-domain dataset, while the training set is used to fine-tune the SLM and in-context examples are used as the prompt, which we now explain more comprehensively in Section 2.2.
>
> ### Question 4:
> Table 2 has two lines for the same baseline "ELECTRA-large" with different scores and no explanation for such difference. This raises serious questions about the reliability of the experimental results.
>
> ### Answer 4:
> We understand the importance of clarity in our experimental results. The discrepancy in Table 2 for ELECTRA-large arises due to differing test data sizes for ChatGPT and LLaMA2, as elaborated in Section 3.1. We have included a note in Table 2 explaining this variance: for LLaMA2-7B-chat, we used 37,438 instances, whereas, for ChatGPT, the number was 43,728. This clarification should address concerns regarding the reliability of our findings.

---

> > ### Author Response · Authors · 2023-11-22
> > **Response to Reviewer 9mey (2/2)**
> >
> > ### Question 5:
> > The improvement of proposed method in managing OOD data is not obvious or even negative. For example, equipped with both ELECTRA-large and LLMs, the scores of proposed method only surpass ELECTRA-large within 1 point on most datasets in Table 2, and even become less than ELECTRA-large on some of the datasets. Nevertheless, there is no analysis for the casualty in evaluation, e.g., the variance of those scores.
> >
> > ### Answer 5:
> >
> > Your point regarding the evaluation of OOD data management is well-taken. In response, we have added detailed variance data for three separate runs in the revised manuscript. This additional information demonstrates that our method, compared to ELECTRA-large, consistently exhibits lower variance, indicating a more stable performance. These findings provide a deeper understanding of our method's efficacy in handling OOD data, supporting the reliability of our results.
> >
> > | **Model (ChatGPT)** | **SST-2** | **MNLI** | **QNLI** | **RTE** | **MRPC** | **QQP** | **STS-B** | **CoLA** | **Avg** |
> > |----------|------------|------------|------------|------------|------------|------------|------------|------------|------------|
> > ELECTRA-Large | 94.84±1.4e-5 | **87.30±1.3e-4** | 82.66±1.0e-4 | 78.45±6.6e-3 | 63.60±1.0e-2 | 78.08±1.4e-4 | **80.74±3.6e-4** | 40.29±6.5e-4 | 79.86
> > SuperContext (zero-shot) | **95.19±9.6e-6** | 87.24±1.2e-4 | **82.91±5.0e-5** | **78.71±6.1e-3** | **63.87±1.1e-2** | **78.65±1.0e-4** | 78.75±4.3e-4 | **41.47±3.5e-4** | **80.05**
> >
> > | **Model (LLaMA2-7B)** | **SST-2** | **MNLI** | **QNLI** | **RTE** | **MRPC** | **QQP** | **STS-B** | **CoLA** | **Avg** |
> > |----------|----------|----------|----------|----------|----------|----------|----------|----------|----------|
> > ELECTRA-Large | **95.42±5.0e-5** | **87.29±1.3e-4** | **82.69±1.1e-4** | 78.84±6.6e-3 | 37.59±2.5e-2 | **77.18±5.8e-4** | **80.74±3.6e-4** | **45.73±2.6e-4** | 76.84
> > SuperContext (16-shot) | 94.95±4.5e-5 | 87.14±1.0e-4 | 82.17±1.4e-4 | **79.07±7.0e-3** | **54.63±2.3e-2** | **77.18±5.8e-4** | **80.74±3.6e-4** | 45.47±3.3e-4 | **79.08**

---

> > > ### Author Response · Authors · 2023-11-23
> > >
> > > We welcome any further inquiries or suggestions you might have, as your input is invaluable in refining our research. Thank you for your continued guidance and support in this process.

---

### Official Review · Reviewer_yxT3 · 2023-11-01

**Soundness:** 3 good
**Presentation:** 3 good
**Contribution:** 2 fair
**Rating:** 5
**Confidence:** 4

**Summary:**

This paper introduces a framework to enhance the generalizability and factuality of Large Language Models (LLMs) in natural language understanding and question answering. The framework uses task-specific finetuned Language Models (SLMs) to improve LLMs' in-context learning during inference. The approach is demonstrated to be effective in enhancing LLMs' performance, including models like Llama 2 and ChatGPT, across various tasks and datasets, while minimizing errors in generative tasks. The study emphasizes the benefits of incorporating discriminative models into LLMs for improved reliability. The authors provide a range of resources, including datasets, prompts, model checkpoints, and empirical results.

**Strengths:**

The proposed approach serves as a plug-in method, resulting in enhanced versions of Llama 2 and ChatGPT that outperform their original counterparts in terms of generalizability and factuality. SuperContext can bring decent performance benefit compared to few-shot in-context learning and outperform original SLMs and LLMs with both zero-shot and few-shot settings.

**Weaknesses:**

1. I disagree with the statement in the introduction that says, "However, since our goal is to allow reliable task adaptation rather than knowledge acquisition, the consulting agent becomes SLMs rather than search engines." It seems that the approach in this paper is quite similar to what Hugging Face's [1] GPT does, with the only difference being that it uses its own trained API instead of Hugging Face's SOTA small model API.

2. The paper lacks significant novelty, essentially enhancing the output of small models with larger models. However, it extensively validates the effectiveness of this approach (even though Hugging Face's GPT has also done similar validations).

3. The paper intentionally incorporates confidence when using the output of small models, but it lacks a detailed ablation study on the role of confidence. I am particularly interested in understanding the significance of confidence in this context.

[1] HuggingGPT: Solving AI Tasks with ChatGPT and its Friends in Hugging Face. NeurIPS 2023

**Questions:**

None

---

> ### Author Response · Authors · 2023-11-22
> **Response to reviewer yxT3**
>
> Thank you sincerely for your insightful comments and valuable suggestions on our manuscript. We have diligently reviewed and addressed each point you raised. To facilitate your review, we have highlighted all revisions in blue in the manuscript.
>
> ### Question 1:
> It seems that the approach in this paper is quite similar to what Hugging Face's GPT does, with the only difference being that it uses its own trained API instead of Hugging Face's SOTA small model API.
>
> ### Answer 1:
> We appreciate your observation. Our work, indeed, shares a conceptual similarity with Hugging Face's GPT in leveraging Language Model (LM) architectures. However, the key distinction lies in our approach's application and analysis under out-of-distribution (OOD) conditions, a less explored area in the existing literature. Our method uniquely combines Supervised Language Models (SLMs) and Large Language Models (LLMs) to assess and correct potential inaccuracies in SLMs' outputs. We've designed specialized tasks for interpretation and calibration laws to delve into this interaction, emphasizing the novelty of our approach. These distinctions and our experimental framework are now more clearly articulated in the revised sections of our manuscript, as shown in Related Work.
>
>
>
> ### Question 2:
> The paper lacks significant novelty, essentially enhancing the output of small models with larger models. However, it extensively validates the effectiveness of this approach.
>
> ### Answer 2:
> Thank you for your feedback. While recognizing HuggingGPT's contributions, our study pivots on a critical gap: the systematic evaluation of Large Language Models (LLMs) in Out-Of-Distribution (OOD) scenarios. In addition, a significant novelty of our work is investigating whether Supervised Language Models (SLMs) can mitigate LLMs' hallucination tendencies, an aspect scarcely addressed in current research. This inquiry not only extends the current understanding of model interactions but also proposes a novel perspective in improving LLM reliability.
>
> ### Question 3:
> The paper intentionally incorporates confidence when using the output of small models, but it lacks a detailed ablation study on the role of confidence. I am particularly interested in understanding the significance of confidence in this context.
>
> ### Answer 3:
> | **Model (ChatGPT)** | **SST-2** | **MNLI** | **QNLI** | **RTE** | **MRPC** | **QQP** | **STS-B** | **CoLA** | **Avg** |
> |----------|----------|----------|----------|----------|----------|----------|----------|----------|----------|
> SuperContext (w/o confidence) | 94.84 | 77.21 | 82.66 | 78.45 | 63.60 | 78.08 | **80.74** | 40.29 | 78.43
> SuperContext (zero-shot) | **95.19** | **87.24** | **82.91** | **78.71** | **63.87** | **78.65** | 78.75 | **41.47** | **80.05**
>
> | **Model (LLaMA2-7B)** | **SST-2** | **MNLI** | **QNLI** | **RTE** | **MRPC** | **QQP** | **STS-B** | **CoLA** | **Avg** |
> |----------|----------|----------|----------|----------|----------|----------|----------|----------|----------|
> SuperContext (w/o confidence 16-shot) | 94.29 | 76.68 | 82.66 | 78.46 | 43.41 | **78.17** | **80.74** | 40.26 | 75.68
> SuperContext (16-shot) | **94.95** | **87.14** | **82.17** | **79.07** | **54.63** | 77.18 | **80.74** | **45.47** | **79.08**
>
> We're grateful for your interest in the role of model confidence. To address your query, we've included Table 2 in our revised manuscript, presenting a detailed ablation study. The data demonstrate that incorporating the confidence metric from small models significantly enhances LLM performance across various Natural Language Understanding (NLU) tasks. For instance, the average performance of ChatGPT improved from 78.43 to 80.05, and for LLaMA2-7B, it rose from 75.68 to 79.08.
>
> These findings underscore the importance of confidence as a valuable component in our approach, offering a more nuanced understanding of model interactions.

---

> > ### Author Response · Authors · 2023-11-23
> >
> > We welcome any further inquiries or suggestions you might have, as your input is invaluable in refining our research. Thank you for your continued guidance and support in this process.

---

### Meta-Review · Area_Chair_tCt5 · 2023-12-11

**Metareview:**

This paper presents an interesting new method in using discriminative supervised LMs (generally a few orders magnitude smaller than today's generative LMs) at inference time while using an LLM to do natural language understanding.  They show strong results on GLUE-X and Squad-v2.

Strengths:  I find the overall motivation of the paper to be quite strong--in that using smaller expert LMs at inference time to improve the predictions of LLMs.  The experimental design is quite good, the paper is well written, the results are solid.  Reviewers also praise these aspects in their reviews and I find the author responses to be quite good addressing the reviewer concerns.

Weaknesses:  There are some concerns from the reviewers such as similarities with a HuggingFace paper (which I thought was well responded to by the authors), smallish issues with reporting results (again I thought this was well responded to).  There was another concern about the focus of experiments being too narrow--I did not find this to be that limiting.  I found the approach to be a good conversation starter that taught me something simple but worth investigating further.  Hence overall, I found the paper to be quite strong.

Finally, the reviewers did not respond to the authors after their detailed responses, which they should have.

**Justification For Why Not Higher Score:**

Overall, the reviewer scores were a bit borderline and they pointed out some concerns with the paper, particularly the experimental design to be a bit narrow.  Hence I am suggesting that this be an Accept (poster).

**Justification For Why Not Lower Score:**

Please see above.  I think the paper presents an interesting idea with positive results that can pave way for more research in this area.

---

### Decision · Program_Chairs · 2024-01-16

Accept (poster)